# Determinants of Refugee and Migrant Health Status in 10 European Countries: The Mig-HealthCare Project

**DOI:** 10.3390/ijerph17176353

**Published:** 2020-08-31

**Authors:** Elena Riza, Pania Karnaki, Alejandro Gil-Salmerón, Konstantina Zota, Maxwell Ho, Maria Petropoulou, Konstantinos Katsas, Jorge Garcés-Ferrer, Athena Linos

**Affiliations:** 1Department of Hygiene, Epidemiology & Medical Statistics, Medical School, National and Kapodistrian University of Athens, 11527 Athens, Greece; eriza@med.uoa.gr (E.R.); a.linos@prolepsis.gr (A.L.); 2Institute of Preventive Medicine, Environmental and Occupational Health, Prolepsis, 15121 Marousi, Greece; p.karnaki@prolepsis.gr (P.K.); d.zota@prolepsis.gr (K.Z.); k.katsas@prolepsis.gr (K.K.); 3Polibienestar Research Institute, University of Valencia, 46022 Valencia, Spain; jordi.garces@uv.es; 4 Faculty of Arts and Sciences, Harvard College, Cambridge, MA 02138, USA; maxwellho@college.harvard.edu; 5Evidence Synthesis Methods Team, Department of Primary Education, University of Ioannina, 45110 Ioannina, Greece; m.petropoulou.a@gmail.com; 6Department of Public Health, School of Health Sciences, University of Thessaly, 43100 Karditsa, Greece

**Keywords:** migrants, refugees, health needs, mental health, chronic diseases

## Abstract

In this study, we collect and synthesize information on the health status of the refugee/migrant population in ten European countries in order to map refugee/migrant health needs. With this information, we identify areas of intervention and healthcare system strengthening to provide the basis for future health planning and effective healthcare provision to migrants, asylum-seekers and refugees in the European Union (EU). Methods: 1407 migrants in ten European Union countries (consortium members of the Mig-HealthCare project) were surveyed on general health, mental health, and specific diseases using an interviewer-administered questionnaire. Descriptive statistics and multivariable linear regression analyses were conducted to investigate the risk factors on general quality of life for migrants and refugees in the EU. Results: Mean age was 31.9 (±11.05) years and 889 (63.1%) participants were males. The majority came from Syria, Afghanistan, Iraq, Nigeria, and Iran. Having a mental health disorder or a chronic disease such as a heart or respiratory disease was associated with worse general health. On the other hand, having permission to stay in the country of interview and being interviewed in the country of final destination was associated with better general health. Access to health care services was fragmented or unavailable for some interviewees because of linguistic, cultural, or administrative barriers. Conclusions: The management of chronic diseases and mental health conditions in European migrants and refugees is a key priority for health service provision. Further efforts should be made to guarantee healthcare access for migrant and refugee populations.

## 1. Introduction

In 2018, 41.6 million (8.12%) people residing in the European Union were born outside of the EU and 2.4 million new immigrants entered the EU-27 from non-EU-27 countries [1]. Heavy migration flows in recent years, due to factors such as the Syrian civil war and the threat of the Islamic State, have contributed to Europe’s “migration crisis” [2].

Given the large influx of vulnerable migrant populations into the EU, the provision of public services like health care is essential to the wellbeing of migrants, refugees, and asylum-seekers. Migrants are often comparatively healthy [3], but their health needs and health profiles are not clearly understood [4,5]. Migrants often face barriers at different levels accessing routine immunizations [6], cancer and chronic disease screening services [7,8]. Migrants, asylum seekers, and refugees also often suffer from psychosocial stressors [9] and high rates of post-traumatic stress disorder, depression, and anxiety, which may go unaddressed by healthcare systems [10]. High rates of mental illness may be partially attributable to lack of social integration, poverty, and barriers in the access to healthcare in host countries, amongst other factors [11,12]. Other social determinants, such as housing conditions, legal status, and discrimination may also contribute to poor migrant health outcomes [13].

After reaching host countries, migrants face difficulties in accessing healthcare. A literature review from the Mig-HealthCare consortium reports that factors such as the migrant’s ability to communicate with health professionals, language, level of health literacy, and awareness of policies and healthcare systems in host countries are common barriers to healthcare (13). Numerous other factors also play a role, such as legal status, entitlement to health care services, issues with continuity of care, treatment follow-up, and fears of detection (for undocumented migrants). There is little research on the use of healthcare services, but numerous infrastructural, organizational, and administrative challenges inhibit effective healthcare provision. In addition, health policies across EU member states vary widely and many fail to address the health of migrants [14]. In this regard, cross-national analysis might play a key role in identifying health inequalities in migrant populations [15].

Recent research focuses on providing equal legal entitlements to migrants, improving living conditions in host countries, designing health policies responsive to migrant needs, identifying the role of primary care, ensuring that migrants are captured in health monitoring, and understanding the need for better comparative research in the EU [16]. Research, however, is still broadly fragmented and incomplete, lacking unified and clear definitions and goals [17]. Variable methodologies and focuses make it difficult to synthesize a holistic understanding of migrants and healthcare in the EU [13].

### Background on the Mig-HealthCare Project

The Mig-HealthCare project is a three year project that was launched in May 2017 [18] partially funded by the European Commission Consumer, Health, Agriculture and Food Executive Agency (CHAFEA). The project’s main scope is to provide evidence-based information and practical guidance to primary healthcare professionals, primarily in the EU Member States, on how to best address the health issues of refugees and migrants. The Mig-HealthCare project especially focuses on community-level interventions in the integration of refugees and migrants.

One of the project’s achievements is a roadmap to effective community-based healthcare models, which aims to improve physical and mental healthcare services, support the inclusion and participation of migrants and refugees in European communities, and reduce health inequalities.

In this paper, we collect and synthesize information on the health status of refugee/migrant populations in ten consortium countries (Austria, Bulgaria, Cyprus, France, Germany, Greece, Italy, Malta, Spain, and Sweden) to map refugee/migrant health needs. We then use this mapping to identify areas of intervention and healthcare system strengthening in order to provide the basis for future health planning and effective healthcare provision. We surveyed migrants, asylum seekers, and refugees in the consortium countries to examine migrants’ general health, psychological distress, and specific disease status. In doing so, we hoped to characterize the unique health profiles of migrants and refugees, while having a comparative framework to understand the commonalities and differences across member countries of the EU. We then used this information to elucidate what systemic and policy changes may be necessary for adequate and effective healthcare to underserved and vulnerable migrant populations in the EU.

## 2. Materials and Methods

### 2.1. Data Collection

All project partners participated in data collection, which lasted from April 2018 to September 2019. The sampling frame for participation in the study/survey was identified by each project partner. Given the diversity of settings in each country, study participants were selected via a snowball technique, a non-probability sampling method which used subjects in each migrant and refugee setting to recruit additional study participants. The ten countries of interview were partners of the Mig-Healthcare project (in alphabetical order): Austria, Bulgaria, Cyprus, France, Germany, Greece, Italy, Malta, Spain, and Sweden. Recruitment of study participants was on a purely voluntary basis from refugees/migrants visiting the points of healthcare delivery services where Mig-HealthCare partners operated (such as health centers in refugee camps, primary health care centers, community centers and NGO clinics).

A short briefing on the survey’s aim was made to the local coordinator of each health provision unit by the Mig-HealthCare project’s partner in each country to guide selection criteria and facilitate interviews (e.g., information on scheduling appointments, availability of interpreters, and privacy).

The protocol of the survey and data collection tools were approved by the ethical committee of the National and Kapodistrian University of Athens Medical School as a representative partner of the coordinating country of the project (Greece). Additional ethical approval was obtained as necessary per each participating partner organization. No identifiable personal data were collected, and a unique identity was assigned to each study participant which was only available to the main researchers. The purpose of the study and the data collection method was clearly explained to each study participant and informed consent was requested before participation.

### 2.2. Study Participants (Selection, Questionnaire Administration, Use of Interpreters, etc.)

#### Inclusion/Exclusion Criteria

A pilot study was conducted in a small sample of 20 eligible study participants to check the clarity, correctness, and order of the questionnaire questions, administration process, and time required for completion. The questionnaire was administered by an interviewer, with the presence of an interpreter when required. The questionnaire was validated and translated into Arabic, Farsi, Dari, Pashto, Somali as well as into the languages of the partner countries (migrant and refugee host countries). All interviewers were trained prior to the initiation of the study.

Eligible study participants were refugees and migrants who were over 18 years of age, had resided in the country of interview for a period of 6 months to 5 years, and were able to both understand the study purposes and give consent for participation in the survey. In the case where study participants could not communicate in the language of the host country of interview, the assistance of an accredited interpreter was sought and provided to complete the questionnaire. Confidentiality agreements were sought and received.

### 2.3. Questionnaire Development and Description

A comprehensive questionnaire was developed for the study, comprising of 60 questions presented in 8 sections: Demographics, Household, Education and Employment, General well-being, Access and interaction with health care services, Health status, Women’s health, and Perceptions about health.

For the General well-being section, an SF-36 questionnaire based on Ware’s 1993 work [18] was designed and translated to each country of interview’s language. The items and dimensions in the SF-36 questionnaire were constructed using the Likert method of summated ratings.

### 2.4. Statistical Analysis

We performed a descriptive analysis for all questionnaire variables. We then conducted multivariable linear regression analyses to investigate the impact of various factors on general health, such as demographic characteristics (country of origin, country of interview), chronic conditions and mental health status, the presence of (co-)morbidity, additional data regarding refugees and migrants’ final destination, the kind of permission to stay in the host country, and health access data.

For chronic conditions, a list of 22 choices was provided to the participant to select from (multiple options possible) following the question: “Do you suffer from any of the following chronic diseases or long-term conditions? (tick all that apply)”.

General health status was assessed based on the SF-36 questionnaire [19], where the respondent was given a set of five component items and was asked to rate them on a 5-category scale, where each scale was assigned with a score. For example, one such component item was “I seem to get sick a little easier than other people” and the scale was: Definitely true (1 point), Mostly true (2 points), Don not know (3 points), Mostly false (4 points), Definitely false (5 points). General health raw score was calculated by summing the scales of its five component items ((1) in general, would you say your health is: good etc.; (2) I seem to get sick a little easier than other people; (3) I am as healthy as anybody I know; (4) I expect my health to get worse; (5) My health is excellent).

Following the established scoring practice provided in the SF-36 questionnaire [18], the transformed raw general health score was derived by converting the sum of the five component items to a value from 0 to 100, where 100 is the best possible health state. The transformed raw score re-calculated as follows [19]:Transformed scale = [(Actual raw score − lowest possible raw score)/Possible raw score range] × 100(1)

The SF-36 questionnaire was evaluated for reliability and validity. After recording questionnaire answers, scoring checks were conducted with Pearson correlation calculations between the general health scale and its five component items to verify that all correlations were positive in direction and substantial in magnitude (0.30 or higher). All variables in the models were tested for collinearity. All analyses were conducted with R statistical software [20]. The level of statistical significance was defined as a = 0.05.

## 3. Results

### 3.1. Sample Characteristics

A total of 1407 migrants and refugees participated in the cross-sectional survey, which was conducted in ten European countries. The general demographics of the sample are presented in Table 1. The mean age was 31.9 (with a standard deviation of 11.05) years and 889 (63.1%) participants were males. The participants migrated from 44 different countries with the majority of the participants (750, 53.9%) migrating from Syria (294, 21.2%), Afghanistan (211, 15.2%), Iraq (127, 9.1%), and Nigeria (118, 8.4%) (Table 1). A table with the absolute and relative frequencies of the participants for each of 44 countries of origin is provided in Appendix A.

### 3.2. Reliability and Validity of SF-36 Questionnaire

The mean general health score was 63.6 (with a standard deviation of 23.4), corresponding to a good general health score. Pearson correlations between general health and its five component items were positive and larger than 0.3, therefore confirming the validity and reliability of general health transformation scale (Appendix A).

### 3.3. Descriptive Statistics

Among the ten European countries of interview, Italy contributed 271 participants (19.26%), Spain 202 participants (14.36%), Greece 255 participants (18.12%) and Bulgaria 225 participants (16.06%). The remaining countries conducted fewer interviews, and Germany conducted only 11 (0.78%) interviews (Table 1).

Table 2 presents the characteristics of participants by country of interview. Fewer males than females were interviewed in Sweden (45%) and Germany (27%). All participants were in their final destination except Greek participants. In Greece, only 36% of migrants reported the host country as their final destination. In France, Austria, Italy, Greece, Spain, and Sweden, over 37% of participants had been granted asylum. In our sample, 358 (29%) were granted asylum, and 619 (46%) had another kind of permission. Syria was a rare country of origin for migrants in France, Italy, Malta, and Spain (<6%). More than 30% of migrants in Bulgaria, France, and Greece came from Afghanistan. Migrants in Austria, Bulgaria, Cyprus, Greece, and Malta were more frequently originally from Iraq (10–21%). In Italy, more than one third was from Nigeria. Regarding health status and country of origin, migrants from Nigeria reported better health status (general health score = 68.6 ± 18.5; *p* < 0.05), compared to migrants from Iraq (general health score = 58.3 ± 25.5) and Afghanistan (general health score = 59.1 ± 26.7). The general health score for migrants from Syria was 63.3 ± 22.3 and for migrants from Iran was 58 ± 27.5.

Table 3 shows the health status of migrants and refugees according to chronic conditions. The most common health problems in migrants and refugees were headaches/migraines, sleep disorders, and mental health issues. The rarest were tuberculosis, cancer, brain stroke, and AIDS/HIV (Table 3).

Table 4 describes migrants’ access to and interaction with health care services. A total of 603 (45.2%) study participants stated that they wanted, but were unable, to use health services during the past 4 months. In total, 464 (35.6%) said that they needed the presence of an interpreter while using health services. In total, 322 (26.7%) believe that they have worse access to the health services compared to the local people.

### 3.4. Multivariable Linear Regression Analyses

We conducted multivariable linear regression models [21] (Models 1–3) investigating the impact of several chronic conditions, physical and mental health status, (co-)morbidity, kind of permission, country of interview, and country of origin in general health (dependent variable) in migrants and refugees (Table 5). Variable selection for regression models (Models 1–3) was initially conceptualized by the perspective of clinical interest and afterwards the selection of the independent variables that provide the best model fit with the stepwise selection method was based on Akaike information criterion (AIC) [22,23]. We conducted a multivariable linear regression analysis to investigate the impact of various chronic conditions on general health (Model 1, Table 5). Male migrants scored higher in general health compared to females (Models 1–3; *p* < 0.05). Model 1 provided that general health-related quality of life is reduced by 15.13 (95% CI 22.89, −7.37) for participants with heart-related chronic conditions. This reduction in general health-related quality of life for refugees and migrants was similar to those suffering from chronic conditions like musculoskeletal problems (−11.24; 95% CI −16.63, −5.86) and hypertension (−11.91; 95% CI −18.89, −4.94). General health-related quality of life was substantially lower for migrants and refugees suffering from cancer (−24.57; 95% CI −42.04, −7.11) and tuberculosis (−20.6; 95% CI −34.8, −6.41). There was no evidence for an association between general health and diabetes, brain stroke, and AIDS/HIV (Model 1, *p* = 0.178, 0.097 and 0.182, respectively).

We conducted a multivariable linear regression analysis to investigate the impact of mental health related chronic conditions and (co-)morbidity on general health (Model 2, Table 5). Model 2 showed that the presence of mental health problems decreased general health-related quality of life by 14.3 (95% CI −19.39, −9.23). Sleep disorders and headaches/migraines were highly prevalent but not significantly associated with general health (*p* = 0.841, 0.324, respectively). We adjusted the covariates: country of interview (10 countries of interview categorized in 9 categories by merging Austria and Germany), country of origin (4 most frequent countries of origin and 1 category for the other countries), having other kind of permission, having permission with asylum (as a separate covariate from other kind of permission), and (co-)morbidity to investigate their impact on general health (Model 3, Table 5). We found no evidence of reduced general health-related quality of life in subjects with one disease (morbidity) (estimate = −4.94; 95% CI −8.18, −1.71) (Model 3, Table 5). Having at least two diseases or chronic conditions (comorbidity) further decreased the general health quality of life (Models 2 and 3; *p* < 0.001).

As presented in Model 3, having a kind of permission to stay in the country of interview was associated with better general health (estimate = 3.23; 95% CI 0.24, 6.21). Syrian migrants reported better general health quality of life than Afghan migrants (estimate = −5.42; 95% CI −10.37, −0.47) and Iraq (estimate = −7.76; 95%CI −12.14, −3.39).

General health score was higher when the country of interview corresponded to the country of final destination for the refugees and migrants of the study (estimate = 5.08; 95% CI 1.88, 8.28). Living in Germany/Austria and Spain indicated higher general health score compared to living in Italy (estimate = 8.66; 95% CI 3.21, 14.11 and 9.84; 95% CI 5.04, 14.65, respectively). Migrants living in Italy had better general health compared to respondents in Bulgaria (estimate = −6.94; 95% CI −13.13, −0.74).

For further analysis, countries of interview were considered in two groups: northern countries (Austria, Germany, Sweden, and France) and southern countries (Italy, Spain, Greece, Cyprus, Bulgaria, Malta) to reflect potential differences in the integration stage of migrants and refugees as in final destination vs transit countries. The same covariates as Model 3 were used. General health did not differ in migrants and refugees between northern and southern countries.

## 4. Discussion

This paper synthesizes self-reported questionnaire data about the health profile and needs of refugee and migrant populations across ten countries in the European Union, connecting health data to general health, various demographics, and social determinants of health. This survey provides aggregated data from a sample of migrants and refugees who have settled in Europe following the large migratory influx after 2015, in an effort to indicate healthcare provision areas that will be in increased demand in coming years. As such, the healthcare services can anticipate these needs and be prepared to respond in an efficient and beneficial way.

The most commonly self-reported health problems were headaches/migraines, mental health problems, sleep disorders, and musculoskeletal illnesses. As in previous studies, chronic diseases such as heart disease, respiratory problems, musculoskeletal complaints, and mental health problems in particular presented a significant burden on the general health of the investigated refugee and migrant population [24]. This is an important finding given the fact that a large proportion of the study population belongs to a working age group of less than 67 years of age and considers the country of interview as their final destination. A previous study by Goodman et al., retrospectively investigating patients at an ambulatory refugee clinic in Germany reported that “the prevalence of chronic disease in this study was relatively low” [25]. However, prevalence may differ from our study simply due to the more specific context in which Goodman et al.’s study occurs. In contrast, in agreement with our findings, a literature review by Lebano et al. (2020) found a high occurrence of chronic disease in some studies on migrant populations. This increased morbidity of chronic diseases and mental health problems identified in this study suggests that inclusive healthcare provision for migrants and refugees will be essential for their health needs [13].

Heart and respiratory diseases are also fairly prevalent in our sample and are associated with a burden on general health. Literature on migrant health in various EU countries often reports that respiratory infections are among the most common infectious diseases, though the exact prevalence varies [25,26,27].

Our results highlight the prevalence and impact of mental illness on migrant health. Numerous studies on migrants and refugees have also previously found elevated levels of mental health problems, in particular post-traumatic stress disorder, depression, and anxiety [11,12,13]. Some of these studies attribute the higher risk of mental health disorders to trauma experienced in the immigration journey, as well as poor social integration, poverty, limited healthcare access, and discrimination in the host country. A critical review by Hynie (2018) emphasizes post-migration factors such as income, employment, and housing as important determinants for mental health and recovery from previous trauma [28]. Hou et al. (2019), in a meta-analysis, find that subjective and interpersonal daily stressors after settling are associated with PTSD and anxiety with mental health [29].

We found that social determinants of health, such as legal status, were associated with general health. Having permissions apart from asylum improves general health. This finding complements the relationship reported in our study between improved general health and country of destination, highlighting the positive effect of safety and stability on perceived health. A literature review by Rousseau and Frounfelker (2018) and a critical review by Hynie (2018) agree that uncertain legal status and the asylum-seeking process might be psychosocial stressors associated with an increased risk of depression and anxiety disorders [12,28]. General health differed across the EU countries as migrants and refugees living in Germany, Austria and Spain indicated higher general health score compared to respondents living in Italy who scored better compared to respondents in Bulgaria. However, when general health was compared between northern (usually final destination countries) and southern countries (usually transit countries), no differences were observed. Other research has noted that migrants are disadvantaged in the European healthcare process, and more research is needed to improve health inequalities and health systems at the international level [30].

Another important finding from our study is that female migrants were more likely to report poorer health, as was also reported in a Greek study by Stathopoulou et al. (2018) [31]. This suggests that future heath care service planning will need to closely address the often-overlooked issues of maternal care and gynecological health.

The present survey highlights perceived health needs and provides evidence on the utilization of health services in a sample of refugees and migrants in 10 European countries. These findings are essential to ensuring the provision of appropriate and efficient healthcare for these populations.

According to the World Health Organization (WHO), the right of health and access to healthcare should be universal and not directed by any conditions such as nationality or legal status. However, in a recent WHO report on migrant health in the European region [32], data show that there is large variation in the provision of healthcare service to refugees and migrants based on legal status (ranging from emergency service use only to unconditional care).

Our findings may have significant policy implications, highlighting areas of unmet need which need to be addressed by European health systems. Actions such as improving communication between health professionals and refugees/migrants, restructuring and reorganizing healthcare systems, and reconsidering formal barriers to healthcare access (e.g., legal, financial) will help ameliorate the unmet health needs in migrant and refugee populations.

## 5. Strengths and Limitations

Strengths of this study include a comprehensive questionnaire which investigated numerous indicators. Migrant populations in ten EU Member States were represented and questionnaires were administered in native and in host country languages. The range of data and populations investigated allows for comparative and intersectional analyses across many dimensions.

Limitations of this study include questionnaires relying on self-reported information. Furthermore, study participants varied greatly in country of origin, duration of stay in country of interview, and integration phase. The use of interpreters may have introduced information bias, and cultural barriers in female representation in the survey for some countries (such as Afghanistan) may have biased their responses. Furthermore, study participants were not randomly selected as it was impossible to define a sampling frame in any of the settings, primarily due to the high mobility of the refugee/migrant populations. As a result, the recruitment of participants was based on a voluntary basis, which may have introduced some degree of selection bias for people who had health issues and who had experience with the host country’s healthcare service. However, as the aim of the study was to provide a mapping of the health needs of the migrant/refugee population in the project’s EU countries, the internal validity of the findings was not affected. Finally, as this is a cross-sectional study which relies on a non-random sample, temporal relationships cannot be established.

## 6. Conclusions

The results of this survey allowed for the mapping of health needs of refugees and migrants in 10 European countries. The findings of the present study may be useful in future health policy planning that will facilitate effective healthcare provision, especially in the management of chronic diseases and of mental health conditions. Furthermore, general health perception is associated with legal status of residence in the host country, as well as access to healthcare services with the provision of linguistic support to ensure effective communication.

## Figures and Tables

**Table 1 ijerph-17-06353-t001:** Characteristics of the total sample of migrants and refugees (N = 1407).

	*N* (*%*)
**Country of Interview**	
Austria	126 (8.96)
Bulgaria	226 (16.06)
Cyprus	110 (7.82)
France	64 (4.55)
Germany	11 (0.78)
Greece	255 (18.12)
Italy	271 (19.26)
Malta	38 (2.7)
Spain	202 (14.36)
Sweden	104 (7.39)
**Five Most Frequent Countries of Origin**	
Syria	294 (21.15)
Afghanistan	211 (15.18)
Iraq	127 (9.14)
Nigeria	118 (8.49)
Iran	48 (3.45)
Final Destination (Yes)	979 (72.73)
Asylum (Yes)	358 (28.94)
Other kind of permission (Yes)	619 (46.37)
	**Mean ± Standard Deviation**
Age (years)	31.98 ± 11.05
BMI (kg/m^2^)	24.79 ± 4.42
Education (years)	9.1 ± 5

**Table 2 ijerph-17-06353-t002:** Characteristics of migrants and refugees by country of interview.

	Austria (N = 126)	Bulgaria (N = 226)	Cyprus (N = 110)	France (N = 64)	Germany (N = 11)	Greece (N = 255)	Italy (N = 271)	Malta (N = 38)	Spain (N = 202)	Sweden (N = 104)
**Country of Origin**										
Syria	31(24.60)	74(32.74)	26(23.64)	2(3.13)	9(81.82)	105(41.18)	2(0.74)	2(5.26)	2(0.99)	41(39.42)
Afghanistan	10 (7.94)	74(32.74)	0(0)	22(34.38)	0(0)	91(35.69)	1(0.37)	2(5.26)	2(0.99)	9(8.65)
Iraq	26(20.63)	36(15.93)	21(19.09)	0(0)	0(0)	32(12.55)	0(0)	5(13.16)	0(0)	7(6.73)
Nigeria	9(7.14)	1(0.44)	4(3.64)	0(0)	0(0)	0(0)	97(35.79)	3(7.89)	1(0.5)	3(2.88)
Iran	6(4.76)	21(9.29)	0(0)	0(0)	0(0)	17(6.67)	0(0)	0(0)	0(0)	4(3.85)
Other	44 (34.92)	20 (8.85)	59(53.64)	40 (62.50)	2 (18.18)	10(3.92)	171(63.10)	26(68.42)	197(97.52)	40(38.46)
Males (%)	63 (50)	142 (62.83)	65 (59.09)	45 (70.31)	3 (27.27)	149 (58.43)	236 (87.08)	25 (65.79)	114 (56.44)	47 (45.19)
Final Destination (%Yes)	92 (74.19)	138 (61.61)	74 (68.52)	44 (91.67)	11 (100)	84 (36.21)	246 (93.18)	16 (51.61)	180 (89.11)	94 (92.16)
Asylum (%Yes)	54 (43.20)	30 (23.26)	21 (20.39)	27 (65.85)	3 (27.27)	85 (37.61)	41 (15.47)	8 (23.53)	47 (23.62)	42 (40.38)
Other permission (%Yes)	65 (52.85)	1 (0.44)	56 (54.90)	19 (43.18)	8 (80)	101 (44.69)	211 (79.32)	30 (90.91)	72 (35.82)	56 (53.85)
Age (years)	31.63 ± 9.82	31.57 ± 12.01	34.20 ± 10.43	28.92 ± 7.72	44.64 ± 16.64	34.88 ± 10.20	25.27 ± 5.76	34.11 ± 11.08	35.38 ± 12.86	35.32 ± 11.43
BMI (kg/m^2^)	24.42 ± 4.11	24.79 ± 4.62	26.29 ± 4.10	24.90 ± 5.83	26.53 ± 5.21	25.03 ± 5.20	23.79 ± 2.98	24.40 ± 3.77	24.90 ± 4.10	25.39 ± 5.30

**Table 3 ijerph-17-06353-t003:** Health status of migrants and refugees.

	*N* (%)
Headaches/Migraines	179 (12.72)
Presence of mental health problem	108 (7.68)
Sleep disorders	95 (6.75)
Illness related to bone and muscle	89 (6.33)
Gastrointestinal disease	82 (5.83)
Respiratory disease	72 (5.12)
Diabetes	59 (4.19)
Hypertension	59 (4.19)
Chronic problems from Injury/accidents	51 (3.63)
Heart disease	44 (3.13)
Kidney disease	35 (2.49)
Tuberculosis	9 (0.64)
Cancer	6 (0.43)
AIDS/HIV	5 (0.36)
Brain stroke	4 (0.29)
General Health score (Mean ± Standard deviation)	63.6 ± 23.4

**Table 4 ijerph-17-06353-t004:** Migrant access to and interaction with health care services.

	*N* (*%*)
Inability to access health care services during the last 6 months.	603 (45.2)
Need for a translator during their interactions with healthcare services:	
*Never*	348 (26.7)
*Few times*	237 (18.2)
*Most times*	253 (19.4)
*Always*	464 (35.6)
Did you need to take medication and were not able to?	
*Not needed*	430 (38.5)
*Needed and had access*	541 (48.5)
*Needed and was not able to take it*	145 (13.0)
Do you believe that you have worse access to health services compared to local people? (Yes)	322 (26.7)

**Table 5 ijerph-17-06353-t005:** Multivariable linear regression models (Models 1–3) investigating the impact of several chronic conditions, physical and mental health status, (co-)morbidity, kind of permission, country of interview, and country of origin on general health (dependent variable) in migrants and refugees.

Independent Variable	Estimate	*p*-Value	95% Confidence Interval
**Model 1** (N = 1129)			
Age (per year)	−0.06	0.312	(−0.18, 0.06)
Gender (males)	4.36	**<0.001**	(1.77, 6.95)
Presence of mental health problem	−15.45	**<0.001**	(−20.28, −10.63)
Heart disease	−15.13	**<0.001**	(−22.89, −7.37)
Illness related to bone and muscle	−11.24	**<0.001**	(−16.63, −5.86)
Hypertension	−11.91	**<0.001**	(−18.89, −4.94)
Gastrointestinal disease	−7.59	**<0.001**	(−12.78, −2.41)
Respiratory disease	−8.35	**0.003**	(−13.85, −2.84)
Cancer	−24.57	**0.006**	(−42.04, −7.11)
Tuberculosis	−20.6	**0.005**	(−34.8, −6.41)
Kidney disease	−8.87	**0.041**	(−17.39, −0.35)
**Model 2** (N = 1129)			
Age (per year)	−0.1	0.119	(−0.22, 0.02)
Gender (males)	4.2	**0.002**	(1.56, 6.83)
Presence of mental health problem	−14.31	**<0.001**	(−19.39, -9.23)
Sleep disorders	0.54	0.841	(−4.78, 5.87)
Headaches/Migraines	−1.97	0.324	(−5.88, 1.94)
Having one disease or chronic condition (morbidity)	−2.74	0.085	(−5.86, 0.38)
Having at least two diseases or chronic conditions (comorbidity)	−16.65	**<0.001**	(−20.4, −12.9)
**Model 3** (N = 1054)			
Age (per year)	−0.16	**0.014**	(−0.29, −0.03)
Gender (males)	5.92	**<0.001**	(3.16, 8.67)
Germany or Austria as country of interview^+^	8.66	**0.002**	(3.21, 14.11)
Bulgaria as country of interview^+^	−6.94	**0.028**	(−13.13, −0.74)
Spain as country of interview^+^	9.84	**<0.001**	(5.04, 14.65)
Afghanistan as country of origin^+^	−5.42	**0.032**	(−10.37, −0.47)
Iraq as country of origin^+^	−7.76	**<0.001**	(−12.14, −3.39)
Nigeria as country of origin^+^	−2.38	0.393	(−7.83, 3.08)
Other country of origin^+^	−5.4	0.095	(−11.76, 0.95)
Having permission with asylum	0.05	0.975	(−2.98, 3.08)
Having other kind of permission	3.23	**0.034**	(0.24, 6.21)
Having one disease or chronic condition (morbidity)	−4.94	**0.003**	(−8.18, −1.71)
Having at least two diseases or chronic conditions (comorbidity)	−18.15	**<0.001**	(−21.36, −14.93)

^+^ Compared with Italy (reference group for country of interview) and with Syria (reference group for country of origin). Results are not provided for all the included countries of interview and countries of origin. P-values with bold denoted a statistical significance.

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
