# Peer review of "Determinants of Refugee and Migrant Health Status in 10 European Countries: The Mig-HealthCare Project"

_ijerph, 2020, doi:10.3390/ijerph17176353_

Round 1

Reviewer 1 Report

Thank you for the opportunity to read this work. I applaud your efforts to study a very vulnerable population. I’d suggest describing a few more policy implications in the discussion or conclusion.

Lines 53-57 could benefit from some rewording or explaining what you mean by research communication and entitlement, in particular.

Section 2.1 please provide an example of how study participants were recruited by your partner agencies. Where did you find respondents? How were they recruited to participate? Did they receive anything for participation?

L137 How were the chronic conditions and MH status questions asked? Give examples.

L138 “kind of permission” needs to be explained more clearly

L142 You should explain that this is an established scoring practice for the SF 36 using citations.

The methods should include additional description and reasoning for why you selected the variables for Models 1-3.

Table 2 needs some formatting

In Table 5, be sure to clearly state all of your reference groups. Is having asylum a reference group or just not included? Or is that having some kind of permission?

Limitation that needs discussing = the potential for sample bias. This isn’t a representative sample, because that would admittedly be impossible. Yet you may have interviewed migrants/refugees who are generally better off than some others.

L237 “work productive age” is awkward. Use another phrase.

L288 “Considering our results, relevant in this study…” is also awkward. Use alternative wording.

Author Response

Dear Editors and Reviewers,

We would like to express our kind thanks for reviewing our submitted manuscript. We have taken all your suggestions very seriously and we are now re-submitting our work having addressed all your comments. All authors have been involved in the current work and have approved all changes and additions.

We have also run a thorough language check and we hope that you will view our revisions satisfactory.

A detailed presentation of the changes made according to the reviewers’ comments are shown below.

Reviewer 1

Comments and Suggestions for Authors

Comment: Thank you for the opportunity to read this work. I applaud your efforts to study a very vulnerable population. I’d suggest describing a few more policy implications in the discussion or conclusion.

Inserted in Lines 299-013 and 307-311

“The present survey highlights perceived health needs and provides evidence on the utilization of health services in a sample of refugees and migrants in 10 European countries. These findings are essential to ensuring the provision of appropriate and efficient healthcare for these populations.

Our findings may have significant policy implications, highlighting areas of unmet need which need to be addressed by European health systems. Actions such as improving communication between health professionals and refugees/migrants, restructuring and reorganizing healthcare systems, and reconsidering formal barriers to healthcare access (e.g. legal, financial) will help ameliorate the unmet health needs in migrant and refugee populations.”

Comment: Lines 53-57 could benefit from some rewording or explaining what you mean by research communication and entitlement, in particular.

Inserted in Lines 51-56

“After reaching host countries, migrants face difficulties in accessing healthcare. A literature review from the Mig-HealthCare consortium reports that factors like the migrant’s ability to communicate with health professionals, language, level of health literacy, and awareness of policies and healthcare systems in host countries are common barriers to healthcare (13). Numerous other factors also play a role, such as legal status, entitlement to  health care services, issues with continuity of care, treatment follow-up, and fears of detection (for undocumented migrants).”

Comment: Section 2.1 please provide an example of how study participants were recruited by your partner agencies. Where did you find respondents? How were they recruited to participate? Did they receive anything for participation?

Inserted in lines 98-104

“Recruitment of study participants was on a purely voluntary basis from refugees/migrants visiting the points of healthcare delivery services where Mig-HealthCare partners operated (such as health centres in refugee camps, primary health care centres, community centres and NGO clinics).

A short briefing on the survey’s aim was made to the local coordinator of each health provision unit by the Mig-HealthCare project’s partner in each country to guide selection criteria and facilitate interviews (e.g. information on scheduling appointments, availability of interpreters, and privacy).”

Comment: L137 How were the chronic conditions and MH status questions asked? Give examples.

Inserted in Lines 140-150

“For chronic conditions, a list of 22 choices was provided to the participant to select from (multiple options possible) following the question: “Do you suffer from any of the following chronic diseases or long term conditions? (tick all that apply)”.

General health status was assessed based on the SF-36 questionnaire (18), where the respondent was given a set of five component items and was asked to rate them on a 5-category scale, where each scale was assigned with a score. For example, one such component item was “I seem to get sick a little easier than other people” and the scale was: Definitely true (1 point), Mostly true (2 points), Don’t know (3 points), Mostly false (4 points), Definitely false (5 points). General health raw score was calculated by summing the scales of its five component items ((1) in general, would you say your health is: good etc.; (2) I seem to get sick a little easier than other people; (3) I am as healthy as anybody I know; (4) I expect my health to get worse; (5) My health is excellent).”

Comment: L138 “kind of permission” needs to be explained more clearly

Inserted in line 139:  “….kind of permission to stay in the host country”.

Comment: L142 You should explain that this is an established scoring practice for the SF 36 using citations.

Inserted in Lines 151-154 We have now revised the sentence with “Following the established scoring practice provided in SF-36 questionnaire (18), the transformed raw general health score was derived by converting the sum of the five component items to a value from 0-100, where 100 is the best possible health state. The transformed raw score re-calculated as follows (18):”.We also added more information “General health status was assessed based on the SF-36 questionnaire (18)….. My health is excellent).”

Comment: The methods should include additional description and reasoning for why you selected the variables for Models 1-3.

We have now provided an additional description for regression models and the variable selection procedure in Lines 208-215.

We added “We conducted multivariable linear regression models Citation: Chambers, J. M. (1992) Linear models. Chapter 4 of Statistical Models in S eds J. M. Chambers and T. J. Hastie, Wadsworth & Brooks/Cole (Models 1-3) investigating the impact of several chronic conditions, physical and mental health status, (co-)morbidity, kind of permission, country of the interview, and country of origin in general health (dependent variable) in migrants and refugees (Table 5). Variable selection for regression models (Models 1-3) was initially conceptualized by the perspective of clinical interest and afterwards statistical methods provided by choosing the independent variables that provide the best model fit with the stepwise selection method based on Akaike information criterion (AIC) Citations: Hastie, T. J. and Pregibon, D. (1992) Generalized linear models. Chapter 6 of Statistical Models in S eds J. M. Chambers and T. J. Hastie, Wadsworth & Brooks/Cole.” and Venables, W. N. and Ripley, B. D. (2002) Modern Applied Statistics with S. New York: Springer (4th ed)

Comment: Table 2 needs some formatting

Thank you for noticing. We have now re-formatted Table 2

Comment: In Table 5, be sure to clearly state all of your reference groups. Is having asylum a reference group or just not included? Or is that having some kind of permission?

No reference category provided in regression models 1 and 2. For all categorical variables, it is obvious the reference category. For example, for cancer in model 1 the estimate is -24.57 compared with non-cancer. Having asylum is included in Model 3 and it now provided in Table 5. We have now made clear that the reference group for the country of origin Syria and the reference group for the country of the interview is Italy (model 3, Table 5)

Comment: Limitation that needs discussing = the potential for sample bias. This isn’t a representative sample, because that would admittedly be impossible. Yet you may have interviewed migrants/refugees who are generally better off than some others.

Inserted in lines 321-327

Furthermore, study participants were not randomly selected as it was impossible to define a sampling frame in any of the settings, primarily due to the high mobility of the refugee/migrant populations. As a result, the recruitment of participants was based on a voluntary basis, which may have introduced some degree of selection bias for people who had health issues and who had experience with the host country’s healthcare service. However, as the aim of the study was to provide a mapping of the health needs of the migrant/refugee population in the project’s EU countries, the internal validity of the findings was not affected.

Comment: L237 “work productive age” is awkward. Use another phrase.

Inserted in line 262 “belongs to a working age group”

Note for the reviewer: we used the term “working age group” which is a statement provided by EUROSTAT https://ec.europa.eu/eurostat/statistics-explained/index.php/Population_structure_and_ageing#:~:text=The%20population%20of%20the%20EU,for%2064.7%20%25%20of%20the%20population. )

Comment: L288 “Considering our results, relevant in this study…” is also awkward. Use alternative wording

Inserted in lines 331-333

The findings of the present study may be useful in future health policy planning that will facilitate effective healthcare provision, especially in the management of chronic diseases and of mental health conditions.

Reviewer 2 Report

This is an interesting study about the health status of refugee and migrant.

The authors need to list all the countries of origin in the supplemental files to give the readers a general idea of how many people are from each of the 44 different countries. Is there a correlation between the countries of origin and health status?

The authors need to list the SF-36 questionnaire and how they were scored.

Are there any differences of healthcare access in different countries?

The supplementary tables are the same as main tables.

Line 144, mistyping. “After recording questionnaire answers, After items have been recorded and scoring checks were conducted with Pearson ….”

Author Response

Reviewer 2

Comments and Suggestions for Authors

This is an interesting study about the health status of refugee and migrant.

Comment: The authors need to list all the countries of origin in the supplemental files to give the readers a general idea of how many people are from each of the 44 different countries.

We have now provided descriptive statistics for participants regarding all the countries of origin in the supplementary files (Supplementary Table 1). We added in manuscript in Lines 168-169 “A table with absolute and relative frequencies of the participants for each of 44 countries of origin is provided in Supplementary Table 1” 

Comment: Is there a correlation between the countries of origin and health status?

We added in lines 190-194 “Regarding health status and country of origin, migrants from Nigeria reported better health status (general health score=68.6±18.5; p<0.05) compared to Iraq (general health score=58.3±25.5) and Afghanistan (general health score=59.1±26.7). General health score for migrants from Syrians was scored with 63.3±22.3 and migrants from Iran with 58±27.5.”

Comment: The authors need to list the SF-36 questionnaire and how they were scored.

We have now added in Lines 143-150 “General health status was assessed based on the SF-36 questionnaire, where the respondent was given a set of five component items and was asked to rate them on a 5-category scale’…..

Transformed scale=[Actual raw score-lowest possible raw score/Possible raw score range]×100”

(1)

Comment: Are there any differences of healthcare access in different countries?

Inserted in lines 302-306

According to the World Health Organization (WHO), the right of health and access to healthcare should be universal and not directed by any conditions such as nationality or legal status. However, in a recent WHO report on migrant health in the European region (28), data shows that there is large variation in the provision of healthcare service to refugees and migrants based on legal status (ranging from emergency service use only to unconditional care).

New Reference (28): WHO Report on the health of refugees and migrants in the WHO European Region: no public health without refugee and migrant health (2018) ISBN 978 92 890 5384 6 https://www.euro.who.int/en/publications/abstracts/report-on-the-health-of-refugees-and-migrants-in-the-who-european-region-no-public-health-without-refugee-and-migrant-health-2018

Comment: The supplementary tables are the same as main tables.

Thank you for this useful comment. All tables remain as main tables except Table 6. We have now moved Table 6 to supplementary.

Comment: Line 144, mistyping. “After recording questionnaire answers, After items have been recorded and scoring checks were conducted with Pearson ….”

Thank you for noticing that. We have now revised this in lines 155-157 with “After recording questionnaire answers, scoring checks were conducted with Pearson…”

Round 2

Reviewer 1 Report

I don’t have many other edits at this time as you adeptly addressed my prior comments. Only a few edits remain and I’d say this is ready to be published. Thank you for the opportunity to review this work again.

L217-8 Why don’t you provide the estimates for gender and age? You refer to gender results in the discussion (L301) but haven’t presented them. I would lean toward including the gender and age estimates in all of the models, especially gender, whether or not they were significant. At a minimum include the gender estimates if you’re going to discuss them in the discussion.

L297-8 states “Moreover, our analysis did not reveal notable differences in self-reported general health across migrant populations in different EU countries.” The results in model 5 seem to suggest otherwise. Migrants in Germany, Austria, or Spain fare better than those in Bulgaria or Italy.

Author Response

Dear Editors and Reviewers,

We would like to thank Reviewer 1 for the two additional comments which we have addressed as indicated below.

Comment 1

L217-8 Why don’t you provide the estimates for gender and age? You refer to gender results in the discussion (L301) but haven’t presented them. I would lean toward including the gender and age estimates in all of the models, especially gender, whether or not they were significant. At a minimum include the gender estimates if you’re going to discuss them in the discussion.

Reply

Inserted in Lines 218-221

“We conducted a multivariable linear regression analysis to investigate the impact of various chronic conditions on general health (Model 1, Table 5). Male migrants scored higher in general health compared to females (Models 1-3; p<0.05)”.

Inserted in Line 228 (Table 5) we included the age and gender estimates in every model.

Comment 2

L297-8 states “Moreover, our analysis did not reveal notable differences in self-reported general health across migrant populations in different EU countries.” The results in model 5 seem to suggest otherwise. Migrants in Germany, Austria, or Spain fare better than those in Bulgaria or Italy."

Reply

Inserted in Lines 251-254

“Living in Germany/Austria and Spain indicated higher general health score compared to living in Italy (estimate=8.66; 95%CI 3.21, 14.11 and 9.84; 95%CI 5.04, 14.65 respectively). Migrants living in Italy had better general health compared to respondents in Bulgaria (estimate=-6.94; 95%CI -13.13, -0.74).”

Inserted in Lines 303-309

“General health differed across the EU countries as migrants and refugees living in Germany, Austria and Spain indicated higher general health score compared to respondents living in Italy who they scored better compared to respondents in Bulgaria. However, when general health was compared between northern (usually final destination countries) and southern countries (usually transit countries), no differences were observed. ”
